



# Weather radar rainfall data in urban hydrology

Søren Thorndahl[1], Thomas Einfalt[2], Patrick Willems[3], Jesper Ellerbæk Nielsen[1], Marie-Claire ten Veldhuis[4], Karsten Arnbjerg-Nielsen[5], Michael R. Rasmussen[1], Peter Molnar[6]

[1]Department of Civil Engineering, Aalborg University, Aalborg, 9220, Denmark
[2]hydro & meteo GmbH & Co KG, Lübeck, 23552, Germany
[3]Department of Civil Engineering, KU Leuven, Heverlee (Leuven), 3001, Belgium
[4]Department of Water Management, Delft University of Technology, Delft, 2628 CN, Netherlands
[5]Department of Environmental Engineering, Technical University of Denmark, Lyngby, 2800, Denmark
[6]Institute of Environmental Engineering, ETH Zurich, Zurich, 8093, Switzerland

*Correspondence to*: Søren Thorndahl (st@civil.aau.dk)

**Abstract.** Application of weather radar data in urban hydrological applications has evolved significantly during the past decade as an alternative to traditional rainfall observations with rain gauges. Advances in radar hardware, data processing, numerical models, and emerging fields within urban hydrology, necessitate an updated review of the state of the art in radar rainfall for urban hydrological applications.

Three key areas of research have been identified as especially important in application of radar data in urban hydrology, given their significant advances over the past decade: 1) Temporal and spatial resolution of rainfall data required for different hydrological applications, 2) Rainfall estimation, radar data adjustment and data quality, and 3) Nowcasting of radar rainfall and real-time applications. Based on these three fields of research, the paper provides recommendations based on an updated overview of shortcomings, gains, and novel developments in relation to urban hydrological applications.

The paper reviews how the focus in urban hydrology as a field of research has shifted over the last decade to fields such as urban resilience to hydrological extremes, climate change impacts, and on-line warning/prediction systems. It is discussed how radar rainfall data can contribute to existing hydrological fields and add value to the aforementioned emerging fields in current and future applications.

## 1 Introduction

In 2003 the International Group on Urban Rainfall (IGUR) under the IWA/IAHR Joint Committee on Urban Drainage initiated joined work on the status and development on using radar rainfall data within in the context of urban drainage. This led to a review paper entitled: "*Towards a roadmap for use of radar rainfall data in urban drainage*" which was published in *Journal of Hydrology* by Einfalt et al. (2004). The paper highlighted the state of the art at the time in weather radar hardware and data processing, as well as methods and challenges in the application of radar rainfall data in urban drainage.



With rapid developments in radar hardware, signal and data processing, development of new methods for data processing and analysis, development in urban runoff models in order to include more complex hydrological processes, etc., the foundation upon which the original paper was based has significantly changed during the past one and a half decade.

The purpose of the current paper remains the same as Einfalt et al. (2004), i.e. to bridge the gap between current precipitation measurements and suitable precipitation information for operation and design of urban drainage systems. Schilling (1991) and Einfalt et al. (2004) summarized the needs as follows: At least 20 years of recordings without data gaps, a volumetric accuracy of less than 3 %, and a spatio-temporal resolution of 1 km$^2$ and 1 minute, respectively.

The scientific interest within the field is evident from the number of publications for specific search strings in different scientific databases. Figure 1 shows the number of publications registered under the keywords "radar + urban drainage" and "radar + urban hydrology" in the databases *Scopus (Elsevier)* and *Web of Science (Thomson Reuters)* for each year in the period 1980-2015.

Out of a total of 142 published papers in *Scopus* with keywords "*radar + urban drainage*" 37 are from 2004 and earlier and 105 are published after 2004. The corresponding numbers for the keywords "*radar + urban hydrology*" are 56 before 2004 and 85 after 2004. The significant growth covers both an increase in applications of weather radar data in urban hydrology, but also continuous improvement and development of methods, algorithms, instrumentation, etc. for estimating rainfall from radar data.

Table 1 presents the three most cited papers according to *Scopus* and *Web of Science* with the keywords "radar + urban drainage".

Both Schilling (1991) and Einfalt et al. (2004) are review papers which provide recommendations for the use of radar data within urban hydrology. The reason for the many citations should probably be found in a need for guidelines in terms of data resolution, rainfall estimation, and applications. Both papers provide a look into the future of radar rainfall in urban hydrology, emphasize that the application of radar rainfall is under development, and the examples of applications are rather sparse. Einfalt et al. (2004) lists a wide range of hydrological applications which shows a potential for use of radar data. Since 2004, many of such hydrological applications have been implemented and tested in both innovation/research projects or in operation of real systems. This paper will therefore provide an updated overview of some of the listed potential hydrological applications from Einfalt et al. (2004) and their current status in terms of documented applications.

Climate change and consequently increase in extreme rainfall have been a significant catalyst for the development in urban hydrological models over the past decade. There is a need to be able to simulate current and future loads on drainage systems and to fully utilize the capacity of drainage systems in order to accommodate for climate change. Furthermore, integrated hydrological models (e.g. integrated urban drainage, river, and inundation models) have become standard tools e.g. to simulate inundation and flood risks in urban areas. With more detailed and distributed models, follow increased demands for good quality high resolution inputs, which promotes the use of radar rainfall data in urban hydrology.

In addition to higher demands for precise local rainfall data, technological developments in hardware as well as data processing and quality have changed significantly since the publication of the papers in Table 1. There is a need to review



the major technical developments during the past decades, highlight the most groundbreaking applications and cases, and finally provide updated recommendations for the applications of radar rainfall data in urban hydrology.

Despite the progress made over the past decades, many applications are still development or innovation projects within research communities. There is therefore a potential to expand knowledge to the industry and water companies in order to

map possibilities of radar data application and to provide confidence in radar data. With potential users (scientists, water utility companies, consultant companies, authorities) of radar rainfall data within the field of urban hydrology as the main target group, this paper will provide an overview of the state of the art in radar rainfall in urban hydrology and urban drainage.

## 2    Paper outline

Since the first use of radars for precipitation measurement, there has been a rapid development and improvement of weather radar hardware, signal processing, software etc., but the fundamental principles of applying weather radar for precipitation measurements has not changed significantly. We therefore refer to the existing literature on the fundamentals of radar and atmosphere physics e.g. antennas, frequencies, bandwidths, polarization, data correction: e.g. attenuation, clutter removal, and reflectivity-rainfall conversion. These fundamentals are indeed crucial for the quality of rainfall estimation and should

definitely not be disregarded by users of radar rainfall, but they are omitted from the paper since they have been discussed in depth in primers such as Doviak and Zrnić (1993), Collier (1996), Bringi and Chandrasekar (2001), Meischner (2004), Michaelides (2008), and Rinehart (2010). Furthermore, there are pioneering and significant journal papers such as Marshall and Palmer (1945), Austin and Austin (1974), Wilson and Brandes (1979), Smith and Krajewski (1991), Krajewski and Smith (2002), Einfalt et al. (2004), Krajewski et al. (2010), Villarini and Krajewski (2010), and Berne and Krajewski (2013)

which also provide general information on specifications and applications of radar rainfall. Also, VDI (2014) and ISO (2017) have produced a standard on precipitation measurement by radar. In this paper we will focus on new developments in applications of radar in urban hydrology. In this context we have identified three key areas of research which seems to be central in a large majority of the publications within the field of radar rainfall application in urban hydrology. The three key research areas are:

-    Temporal and spatial resolution of radar data (Section 3.1)

       -    Rainfall estimation, radar data adjustment and quality (Section 3.2)

       -    Nowcasting of radar rainfall (Section 3.3)

We will approach these key areas from two sides. Initially, in Section 3 we will review state of the art within the three key

areas respectively, focusing on the radar and radar-rainfall related issues. Secondly, we will end each sub-section by reviewing the impacts of each of the key areas on application within urban hydrological modelling.



The second part of the paper will focus on the value of applying radar rainfall in urban hydrology giving examples of current and future applications (Section 4). Finally in Section 5, we will present our subjective views on what is needed and what can be recommended for applications of radar rainfall data in urban hydrology.

## 3 State of the art in radar rainfall estimation for urban hydrological applications

Following Willems (2001), Thorndahl et al. (2008), Schellart et al. (2012b), and others, rainfall input errors are one of the most important sources of uncertainty in (urban) hydrological models. For example, for a sewer system model in Belgium, it was shown by Willems and Berlamont (1999) that about 20 % of the total uncertainty in the downstream sewer throughflow discharges could be explained by the spatial variability of the rainfall and about 20-25% by the rainfall measurement errors, consisting in their case of rain gauge calibration errors, rainfall intensity resolution errors and errors by wind and local disturbances. For extreme events, e.g. flash flooding, uncertainties related to spatial variability and rainfall measurement errors are expected to be even larger (e.g. Berne et al., 2004; Hossain et al., 2004). Hence there is a need for high quality and high resolution rainfall inputs into urban hydrological models in order to reduce uncertainty in hydrological response.

### 3.1 Temporal and spatial resolution of radar data

Urban hydrology is characterized by fast runoff and short response times on impervious surfaces, thus small time and space scales compared to rural hydrology. Radar rainfall data for urban hydrology is therefore required to resolve these spatial and temporal scales sufficiently. In the following sections, we review and discuss the state of the art with regard to temporal and spatial resolution of radar data.

#### 3.1.1 Temporal resolution

The temporal resolution of radar data is governed by the scanning strategy of the radar. A radar scanning the atmosphere in different elevations to generate a full azimuthal volume scan can take up to several minutes depending on rotational speed and the number of scanning elevations. Radar collects instantaneous samples of rain rates (estimated from reflectivities), unlike rain gauges which accumulate rainfall over a given time interval. Some radars operate with intermediate dedicated Doppler scans for each volume scan, hence doubling the time between two consecutive reflectivity scans. Operational meteorological S-,C-, and X-band usually provide reflectivity scans with a temporal resolution of 5-15 minutes (Table 2) whereas research radars dedicated to high resolution rainfall monitoring in specific areas and specific elevations are reported to provide data resolutions down to 15 sec (e.g. van de Beek et al., 2010; Mishra et al., 2016).

#### 3.1.2 Spatial resolution

The main strength of radars for rainfall estimation is their capability to provide spatially distributed rainfall information. The spatial resolution of radar rainfall data is basically determined by the hardware and physics. The radial resolution (or range



resolution) is a function of the pulse and wave length and can in principle be very small for all radar bandwidths. However, for operational radars the radial resolution is often an indirect function of the radar range due to storage and data transmission restrictions. Thus, each radar scanline is subdivided into a fixed/selected number of range bins, which eventually determines the range resolution of the data. X-band radars with shorter range than C- and S-band radars, are therefore typically operated with a finer radial resolution, e.g. up to a maximum of 500 m. Radial resolution between 3 and 100 m have been documented by e.g. Leijnse et al. (2010), van de Beek et al. (2010), Lengfeld et al. (2014), and Mishra et al. (2016).

The spatial resolution also depend on the azimuthal (or angular) horizontal resolution, which is a function of the beam width determined by the size and design of the antenna. In contrast to the radial resolution the azimuthal resolution will decrease as a function of the radial distance from the radar. Most operational weather radars use symmetrical antennas with a beam width of approx. 1 degree, thus functioning with an azimuthal horizontal resolution close to 1 degree (http://www.eumetnet.eu/opera). As an example a distance of 100 km from the radar will thus lead to a width of the beam of ~1750 m. Small local X-band radars, with non-parabolic antennas typically have larger opening angles between 2 and 3 degrees, but also a smaller range compared to meteorological radars (Pedersen et al., 2010a, 2010b; Nielsen et al., 2012; Thorndahl and Rasmussen, 2012; Goormans and Willems, 2013; Nielsen et al., 2013; Borup et al., 2016).

Typical values for spatial resolutions and maximum ranges for operational radars are provided in Table 2. Generally, operational X-band radars function with both higher spatial and temporal resolution than C- and S-band radars, but there are examples of configurations of C- and S-band radars where high resolution data is derived. A *super-resolution* can be achieved by shortening pulse lengths to obtain higher range resolutions (e.g. Seo and Krajewski; 2010, Sharif and Ogden, 2014; Ochoa Rodriguez et al., 2015) or with adaptive scanning strategies to capture the most intensive part of a storm with a high degree of detail (e.g. Dolan and Rutledge 2010).

Examples of radar reflectivity with four different spatial resolutions covering an area (~12 km x 12 km) over the city of Aalborg, Denmark are shown in Figure 2. The example illustrates the importance of high spatial resolution data in order to capture spatial variability of rainfall over an urban area.

### 3.1.3    Projection of data

Many applications of radar data in urban hydrology favor projected cartesian (gridded) over polar data with decreasing resolution as a function of range. The limit for generating high resolution cartesian data is mainly related to the azimuthal resolution and thus range. Two common methods of data projection are: 1) the Constant Altitude Plan Position Indicator (CAPPI) in which different volume scans are merged in order to generate a radar product with altitude independent of range (however with an inhomogeneous zone where changing from one elevation to the next one), and 2) the Plan Position Indicator (PPI) which applies one scan elevation only thus has an increasing altitude as a function of range.

Due to the curvature of the earth, the refraction of the radar beam through the atmosphere, and wind drift of the raindrops, assigning a radar measurement to a specific point on the surface can be quite challenging. This should also be considered





working with high spatial resolution radar data, since it is not certain that the rain can be allocated with the same accuracy at ground level as in a specific elevation.

### 3.1.4    Advection interpolation (temporal downscaling)

In order to increase the temporal resolution of operational meteorological radar data, especially for urban hydrological applications, some authors have developed methods to interpolate between radar images (Atencia et al., 2011; Jasper-Tönnies and Jessen, 2014; Nielsen et al., 2014; Thorndahl et al., 2014b; Wang et al., 2015a). The governing principle in these downscaling methods is to apply the advection field of the rain, and by resampling in space, convert the spatial resolution into temporal resolution. The methods have been proven to give better local peak estimates of rainfall intensities as well as more accurate accumulated quantitative precipitation estimates in comparison with point ground observations. Jasper-Tönnies and Jessen (2014), Nielsen et al. (2014a); Seo and Krajewski (2015), and Wang et al. (2015a) have successfully converted data with a 5 or 10 minute resolution into a product with 1 minute resolution for use in urban hydrological modelling. The concept of advection interpolation works if the raw radar data are instantaneous. If radar data is averaged (by multiple scans) over a time period, advection interpolation will not be favorable and temporal resolution cannot be increased. In relation to urban hydrological modelling, where very fine temporal resolution indeed is needed for some applications (e.g. down to 1 min), the radar data based on instantaneous sampling are therefore preferable.

Considering the advective nature of rain, it is also clear that advection interpolation yields a better estimate of the area precipitation. Accumulation of instantaneous radar data with e.g. 10 minute sampling rate might result in a "fishbone" pattern.

Commercial radar rainfall products (see section 3.2.4) often provide data which have been temporally accumulated or averaged hence a coarser temporal resolution of data can be found in these products.

### 3.1.5    Impacts of temporal and spatial resolution of radar data in hydrological modelling

In the literature, the impact of spatial and temporal radar data resolution on hydrological model response have been studied intensively (Quirmbach and Schultz, 2002; Berne et al., 2004; Villarini et al., 2010; Emmanuel et al., 2012b; Gires et al., 2012; Liguori et al., 2012; Nielsen et al., 2012; Schellart et al., 2012b; Vieux and Imgarten, 2012; Gires et al., 2013; Lobligeois et al., 2014; Bruni et al., 2015; Gires et al., 2015; Ochoa-Rodriguez et al., 2015; Rafieeinasab et al., 2015; Wang et al., 2015a; Thorndahl et al., 2016). Other than different spatial and temporal resolutions of radar rainfall input data, these studies represent a vast variety of different number of events, severity of events, radar types, catchment sizes, shapes and slopes, catchment imperviousness, models, model scales and resolutions (fully distributed, semi distributed or lumped), model outputs (e.g. peak flows, water levels, volumes, etc.), objective functions for evaluating and comparing results, etc. More general impacts of the spatial and temporal data resolution are therefore difficult to derive since they to a large extent depend on the studied setup. Three different significant issues have been identified related to the requirements for spatial and temporal resolution in runoff response modelling:





1. *For increasing catchment sizes demand for high spatial and temporal resolution radar rainfall data decreases.*

   Schilling (1991) and Einfalt et al. (2004) recommended a minimum temporal resolution of 1-5 min and a minimum spatial resolution of 1 km for the application of radar data in urban hydrology in general. Berne et al. (2004) detailed this to ~1 min/2 km for 10 ha catchments, ~3 min/3 km for 100 ha catchments and ~6 min/4 km for 1000 ha catchments. For even smaller catchments with an area of 1 ha or less, recent studies by Ochoa-Rodriguez et al. (2015) suggest a minimum resolution of 1 min/100 m.

2. *Catchment characteristics and modelled runoff response play an important role in defining the required temporal and spatial radar data resolution.*

   Basically the concentration time of the urban catchment or to a point of interest in the system are of importance and affected by many factors. According to the rational method (Kuichling, 1889) increasing concentration times will lead to greater critical rainfall aggregation levels (in this case coarser temporal resolution), and due to the dependence between temporal and spatial resolutions described above, increasing concentration times will reduce the demands for high spatial resolution. Thus high space-time resolution is required for the simulation of peak runoff response (surcharge, local flooding, etc.) upstream in an urban system, while for the simulation of total catchment runoff or basin storage the requirements on resolution may be reduced (Berne et al., 2004; Bruni et al., 2015; Rafieeinasab et al., 2015).

3. *Storm characteristics (size, movement, shape, lifespan, intensity, etc.) can be important for choice of spatial and temporal resolution.*

   The ability to resolve rainfall adequately in time and space for urban hydrological application depends on the velocity of rainfall fields. By studies of variograms at different temporal aggregation levels and simulation of runoff response, Ochoa-Rodriguez et al. (2015) found a strong interaction between the temporal and spatial resolutions and the impacts on urban runoff response. Berne et al. (2004) suggested a relation between the temporal (t in min.) and spatial (r in km) resolution of: $r = 1.5t^{0.5}$ for Mediterranean rainfall conditions.

   The type and severity of a storm might also set requirements to the space-time resolution. A high-intensity convective thunderstorm with small spatial extent will need a higher resolution in both space and time to be resolved in contrast to a stratiform long-duration storm. This is again related to the runoff response of the system in question. Germann and Joss (2001), Berne et al. (2004), Bruni et al. (2015), and Ochoa-Rodriguez et al. (2015) suggested to apply climatological variograms to characterize the spatial structure of rainfall fields, and to investigate the spatial resolution requirements (given a specific temporal resolution) in order to resolve the spatial structure of rainfall fields in a sufficient way for urban hydrological applications.

## 3.2 Rainfall estimation, radar data adjustment and quality

The use of radar data implies that the data are of good quality. Since there are numerous items such as radar hardware calibration, clutter removal, overshooting/vertical profile correction, etc. (Michelson et al., 2004; Villarini and Krajewski,





2010) which may play a role before radar reflectivity data can be converted into reliable rainfall intensities, a thorough quality check and potential correction are required. Disturbances for a good radar measurement may be undesired reflections off mountains or high towers, air planes, ships, wind turbines, attenuation by heavy rain or hail, snow or melting snow instead of rainfall, anomalous propagation conditions and others. Methods to test for these problems exist, and they are

partly reduced by dual-polarization information from new generation radars. The preprocessing of radar data by meteorological services usually only covers a part of the above points.

Observed radar reflectivity can be converted into rain rates (intensities), but in order to produce valid *quantitative precipitation estimates (QPE)* comparison and adjustment with against ground observations is required. This is most often referred to as radar rainfall adjustment or radar-rain gauge merging and is presented in the following section.  Rain gauges

for adjustment also need to be of high quality. Frequently observed shortcomings of rain gauge data are missing data, time shift (or differently set clocks), clogging of the gauge, data transmission drop outs, gauge calibration errors, local wind effects around gauges leading to measurement errors, or gauge sampling errors (e.g. Ciach, 2003; Villarini et al., 2008b; Gires et al., 2014). In order to avoid random or systematic errors, such effects need to be eliminated before rain gauge data are used to adjust radar rainfall.

### 3.2.1    Reflectivity-rain rate conversion

Radar reflectivity, $Z$ $(mm^6/m^3)$ depends on the *drop size distribution (DSD)* of the target precipitation. Conversion into rain rate, $R$ $(mm/h)$ therefore depends on the size of the individual drops. As documented by numerous authors (e.g. (Marshall and Palmer, 1945; Uijlenhoet and Pomeroy, 2001) the most typical conversion for single polarization radars is to apply a *two-parameter* power-law relationship to describe the relation between rain rate and reflectivity (*Z-R* relationship): $Z=aR^b$.

Since the power-law parameters will vary with the *DSD*, i.e. the type of rain, they will not be constant in time. One solution is to adjust the *Z-R* relationship continuously by use of ground observations. It is however more common to apply a fixed *Z-R* relationship and perform a posteriori bias adjustment (see next section). Whereas traditional *Z-R* conversion has been documented in numerous applications of radar, there are recent advances in the application of dual-polarized radars which enable accurate *QPE* assessment using polarized moments (e.g. Scarchilli et al., 1993; Bringi and Chandrasekar, 2001;

Anagnostou et al., 2004; Anagnostou and Anagnostou, 2008; Bringi et al., 2011; Mishra et al., 2016). Polarization of a radar signal characterizes the orientation of the electric field (both transmitted and received). Dual-polarimetric radars transmit a radar signal alternately in *horizontal (H)* and *vertical (V)* polarization. Depending on the shape of the rain drops, two different signals will be received: reflectivities $Z_{HH}$ and $Z_{VV}$. Additionally, the phase of the horizontally and vertically polarised return signals, $f_{HH}$ and $f_{VV}$, are measured (Illingworth, 2004). Four parameters can be defined based on the

polarimetric measurements: differential reflectivity $Z_{dr}$, linear depolarization ratio $L_{dr}$, co-polar correlation coefficient $r_{co}$ and the specific differential phase $K_{dp}$ (Illingworth, 2004). It has been shown that $K_{dp}$ is proportional to the product of rainwater content and the mass-weighted mean diameter (Bringi and Chandrasekar, 2001) and thus can be used to estimate rainfall rates. The advantage of using $K_{dp}$ for rainfall rate estimation is that it is more sensitive to the raindrop shape, thus rainfall





rate can be estimated from $K_{dp}$ in the case of rain/hail mixture. As soon as the hydrometeors are spherical or quasi-spherical, $K_{dp}$ is about 0°/km (hail, light rain). The advantage of using $K_{dp}$ is also that it is independent of radar calibration and not sensitive to attenuation, an issue of particular importance at X-band frequency. $K_{dp}$ can only be estimated for medium to high rainfall rates (Otto and Russchenberg, 2011).

### 3.2.2   Bias adjustment against ground observations

Many different methods have emerged in the last decade for adjusting rain rates estimated from reflectivities and several profound review papers on different adjustment/merging techniques related to hydrological applications exist (e.g. Goudenhoofdt and Delobbe, 2009; Wang et al., 2013; McKee and Binns, 2016). For specific details we refer to these. Below, we present some of the most widely applied methods.

One of the simplest methods of adjusting radar rainfall data has been proposed by Smith and Krajewski (1991) who introduced the concept of *Mean Field Bias (MFB)* adjustment. The concept is to estimate the ratio between accumulated rainfall in a number of ground observation points (rain gauges) and accumulated radar rainfall in the corresponding points (or grid cells if the radar data is projected onto a cartesian grid). Under the assumption that the radar field has a homogeneous *DSD*, the whole radar field is multiplied by the *MFB* factor. The *MFB* factor should be based on a temporal

integration of data over a period of time in which the *DSD* does not change significantly. If the integration period is too short (e.g. in the range of the temporal resolution of radar data), the bias assessment becomes vulnerable to random errors. On the other hand if the integration period is too long, the adjusted radar rainfall might be inaccurate due to temporal changes in the *DSD* (Krajewski and Smith, 2002). Within urban hydrology most commonly hourly (e.g. Borga et al., 2002; Thorndahl et al., 2014b; Rico-Ramirez et al., 2015; Wang et al., 2015b) or daily (e.g. Seo and Breidenbach, 2002; Wright et al., 2012;

Thorndahl et al., 2014a) *MFB* adjustment is applied.

The optimal temporal integration period or spatial aggregation level is to a large extent dependent on the representativeness of the gauges (gauge network density) to capture the temporal and spatial variability of the rain (e.g. Gires et al., 2014). It is difficult to recommend specific gauge network densities for radar rainfall adjustment since the optimal value will depend on storm type, homogeneity of the rain gauge network, orographic features of the rain, adjustment methods, etc. Generally you

will need a rain gauge network with a higher density for smaller aggregation levels or in other words, the density of the rain gauge network will determine the temporal aggregation level of the radar rain gauge adjustment. McKee and Binns (2016) suggest conducting a sensitivity analysis in order to identify the effect of gauge density on rainfall estimation.

For annual precipitation measurements WMO (2008) recommends 1 per 5750 $km^2$ for plains and 1 per 2500 $km^2$ for mountainous areas. Furthermore, for application in design and management of stormwater systems WMO (2008)

recommends 1 rain gauge per 10-20 $km^2$ for urban areas. For radar rainfall adjustment such density it is not necessary since the radar data will provide information on the spatial and temporal variability of the rainfall. As an example Goudenhoofdt and Delobbe (2016) found no remarkable improvement in the *MFB* assessment of daily rainfall for gauge densities between



per 500 and 1 per 135 km². The relationship between temporal aggregation level, gauge density and spatial interpolation was also studied in detail for kriging based merging methods by Berndt et al. (2014)

As an extension of the *MFB* adjustment, the concept of *conditional MFB* adjustment was proposed by Ciach et al. (2000), Ciach et al. (2007), and Villarini et al. (2008a). The *conditional MFB* adjustment introduces a range (distance) dependent

bias in order to account for rain rate dependent biases especially for convective rainfall with rapidly changing *DSD*. Wright et al. (2014b) demonstrate that especially estimation of large rain rates can be improved significantly by introducing *conditional MFB* adjustment, whereas Thorndahl et al. (2014b) conclude an insignificant effect of *conditional MFB* adjustment for advection interpolated radar data.

### 3.2.3    Spatial variability adjustment

Geostatistical merging of radar and rain gauge data covers another range of methods which are widely applied for *QPE*. The concept here is to merge the spatial variability of the radar rainfall fields into the interpolated rain gauge precipitation fields in order to increase the spatial resolution of this product. The interpolation can be performed by many different spatial interpolation methods e.g. variations of kriging (Krajewski, 1987; Todini, 2001; Sinclair and Pegram, 2005; Haberlandt, 2007; Goudenhoofdt and Delobbe, 2009; Velasco-Forero et al., 2009; He et al., 2011; Berndt et al., 2014; Rabiei and

Haberlandt, 2015) or by *inverse distance weighting* or *Thiessen polygon weighting* (Johnson et al., 1999; Haberlandt, 2007). The kriging based methods rely on variograms for describing the spatial dependence in rainfall fields and are in general more computationally demanding than weighting methods. The latter are therefore often used in real-time operation.

Other methods such as the singularity approach (Wang et al., 2015b) have been proposed in order to overcome problems with spatial smoothing as a results of the variograms in the Kriging based methods. Geostatistical merging and spatially

distributed bias adjustment is mostly applied for radar composites or in mountainous areas with orographic rainfall effects (e.g. Germann et al., 2006; Sideris et al., 2014). Merged rainfall products are described in section 3.2.4.

Another alternative to the optimization and sensitivity approaches of radar-gauge adjustment described above, is to model errors and thereby acknowledge uncertainties in rainfall estimates (e.g. Ciach et al., 2007; Gires et al., 2012; Pegram et al., 2011; Villarini et al., 2014; Rico-Ramirez et al., 2015). It is expected that these uncertainty based methods and development

of rainfall ensembles for hydrological applications will gain more impact in future applications, concurrently with development in probabilistic/ensemble models for urban hydrology.

### 3.2.4    Commercial radar rainfall products

Today, most national meteorological services produce radar rainfall products consisting of radar composites from national radar networks. They provide state of the art corrected CAPPI or PPI products which have been adjusted or merged with rain

gauge network data in order to provide users with best possible rainfall estimates for historical records or in real-time. Examples of products are in Germany: *RADOLAN*, in UK: *NIMROD* and in USA: *NEXRAD*.



These *QPE* products are often provided in a fixed cartesian grid with data summarized over a fixed time period. In some cases only on historical data in hourly or daily precipitation maps but in other cases also fine temporal resolution data are available.

Generating radar composites merging data from two or more radars might be subject to inconsistencies in radar data due to merging of data from different elevations, with different scanning strategies, and using different merging techniques. In application of commercial *QPE* products it is important to be aware of these inconsistencies.

### 3.2.5 Dynamic adjustment in real-time

Operational real-time continuous adjustment of radar rainfall against rain gauges constitutes a challenge in comparison to event based or discontinuous adjustment based on historical data (off-line mode). Nonetheless, for real-time operation of urban hydrological systems, it is crucial to be able to produce valid rainfall estimates in an on-line mode. The real-time adjustment is especially difficult in the beginning of rainfall events with no prior rain gauge data recordings or in situations with large spatial variability of the rainfall. In these cases where rain gauge observations might be sparse and thus subject to domain sampling errors, bias adjustment might be dominated by random factors and can easily result in a erroneous adjustment (Seo et al., 1999; Krajewski and Smith, 2002; Nielsen et al., 2014a). The accuracy of a real-time bias adjustment is thus dependent on the temporal aggregation scale at which the adjustment is performed. The shorter the aggregation scale (e.g hourly or sub-hourly) the larger the risk of erroneous adjustment due to sampling errors and the larger the aggregation scale (e.g. daily or super-daily) the larger the risk of errors due to changes in *DSD* and bias over the aggregation interval. Several authors apply *MFB* adjustment rather than area-based adjustment in real-time operation due to the fact that the latter is more vulnerable to rain gauge sampling errors (Seo et al., 1999; Borga et al., 2000). In order to avoid abrupt changes in bias several authors have suggested to apply algorithms to smooth the bias in time, e.g. using Kalman filtering (Chumchean et al., 2006) or exponential smoothing (Seo and Breidenbach, 2002).

### 3.2.6 Choosing adjustment procedures for hydrological modelling

It is evident, that for hydrological modelling, the most accurate rainfall estimates at ground level are desired. Different adjustment methods and their impacts have been researched in recent studies, e.g. Quirmbach and Schultz (2002), Tilford et al., (2002), Vieux and Bedient, (2004a), Emmanuel et al. (2012a), Gires et al. (2012), Goormans and Willems (2013), Wang et al. (2013), Leonhardt et al. (2014), and Rico-Ramirez et al. (2015). It is difficult to recommend one method of adjustment over another, since it to a large extent depends on the application in question. Instead we have identified some of the key issues related to the requirements of radar rainfall adjustment or radar-rain gauge merging for runoff response modelling in urban areas:

1. *Catchment characteristics are important for the choice of the radar rainfall adjustment method.*

    The choice of the adjustment method depends on the required accuracy the spatially distributed rainfall in the application and the radar rainfall product available. For catchments and spatially homogeneous rainfall events, an





adjustment using rain gauges in or outside the catchment and a fixed *MFB* adjustment might be sufficient to represent rainfall variability. For a large catchment potentially covered by multiple radars, geostatistical merging techniques are required to represent the variability in *DSD* within the study domain and thus more sophisticated techniques might be preferred (see e.g. Wang et al., 2013, 2015b).

*2. Overall model uncertainty might have a significant impact on the urban hydrological model outputs leaving accurate radar rainfall adjustment less crucial.*

Urban hydrological model outputs are subject to uncertainties associated with rainfall inputs as well as representation of hydrological and hydraulic processes, expressed in parameter and model structure uncertainties (e.g. Freni et al., 2008; Thorndahl and Willems, 2008; Thorndahl et al., 2008; Willems, 2008; Dotto et al., 2012). In

cases where parameter uncertainty estimation of such processes dominates runoff response, the rainfall input to urban hydrological model may become less important. Instead of adjusting the radar rainfall individually, some authors have therefore calibrated or optimized hydrological models directly to match runoff response observations without specific adjustment of the rainfall input (Krämer et al., 2005; Ahm et al., 2013; Thorndahl and Rasmussen, 2013; Löwe et al., 2014). However, this is recommended only if parameter or model uncertainties are high and/or

radar rainfall data adjustment is not possible, because it may lead to error compensation with undesired consequences for prediction.

*3. In real-time application, change in storm characteristics might influence the radar rainfall inputs to hydrological models.*

It is of utmost importance that real-time adjustment of radar data reflects the potential changes in *DSD*. In case of

rapid changes e.g. between convective and stratiform precipitation a bias shift might occur. The aggregation time on which a bias (either mean field of spatially varying) adjustment is performed should therefore be able to reflect these changes. This will to a large degree also depend on the density of rain gauges available for adjustment. Required gauge density for an unambiguous adjustment will thus depend on the aggregation level on which the adjustment is performed as well as the storm extent and homogeneity of the storm.

**3.3 Nowcasting of radar rainfall**

Next to the interpolation for urban design, control or scenario simulation applications, temporal extrapolation of radar rainfall fields forms the basis of real-time forecasting and control (e.g. Sharif et al., 2006; Smith et al., 2007; Javier et al., 2007; Achleitner et al., 2009; Einfalt et al., 2009; Liguori et al., 2012; Schellart et al., 2012a; Wang et al., 2012; Thorndahl et al., 2013; Ntegeka et al., 2015). Due to the short response time of the urban drainage system, the short life time and small

spatial size of convective rain cells, urban rainfall forecasts are only reliable for very short lead times (Achleitner et al., 2009; Foresti et al., 2016). Short-term forecasts are called nowcasts and provide input for real-time warning and/or control of urban floods or CSO pollution.





Several generic methods have been developed to nowcast radar data, based on deterministic approaches:, e.g. TREC (Rinehart and Garvey, 1978), CO-TREC (Li et al., 1995), SCIT (Johnson et al., 1998; Mecklenburg et al., 2000), SCOUT (Einfalt et al., 1990) or stochastic approaches: e.g. MAPLE (Turner et al., 2004), STEPS (Bowler et al., 2006). We refer to the individual papers for detailed descriptions of the methods and focus instead on the application of nowcasts within urban hydrological applications here. We have identified three issues which constitute the current major challenges:

1. *Extrapolation of observed radar rainfall has a limited lead time.*

   Despite development of the aforementioned methods, rainfall nowcasting for urban drainage applications is still in its infancy. Albeit rain cells can be extrapolated by radar image extrapolation (e.g. Thorndahl et al., 2013; Löwe et al., 2014) or applying cell tracking (e.g. Sharif et al., 2006; Einfalt et al., 2009; Muñoz et al., 2015), this is often of limited value given the limited duration of rain cells, especially during convective conditions. The quality of an extrapolation-based nowcast depends on the radar range, possible merging of radar networks, resolution, climate zone, and rainfall type. For a standard deterministic nowcast, the lead time varies between less than 30 minutes (small convective cases) to more than two hours (large scale slow moving systems). As a rule of thumb, extrapolation is more difficult with small rainfall cells and for small target areas, less difficult with large rain fields and large target areas.

2. *For reliable nowcasting, stochastic uncertainty should be included.*

   The most promising alternative to simple extrapolation of radar rainfall data is to perturb the deterministic radar extrapolation with stochastic noise to account for the unpredictable rainfall growth and decay processes (Bowler et al., 2006; Germann et al., 2009; Liguori and Rico-Ramirez, 2013). The stochastic noise model aims to describe the nowcast error together with its spatial and temporal correlations. In the Short-Term Ensemble Prediction System (STEPS), this is done by adding stochastic perturbations to the deterministic Lagrangian extrapolation of radar images (Liguori and Rico-Ramirez, 2013). The perturbations moreover aim to reproduce the dynamic scaling of precipitation fields, i.e., the observation that large-scale rainfall structures are more persistent and predictable than small-scale convective cells. STEPS was originally co-developed by the UK Met Office and Australian Bureau of Meteorology, and is currently further customised for urban applications, e.g. in the UK (Liguori et al., 2012; Liguori and Rico-Ramirez, 2012), STEPS-BE for the Belgian version (Foresti et al., 2016). It provides probabilistic ensemble nowcasts. So far, however, these nowcasting systems rely on radar data that is too coarse for urban applications (e.g. 1 km resolution C-band radar data for STEPS-BE).

3. *There is a challenge in combining high resolution radar observations with nowcasts.*

   Future developments will likely involve the use of higher resolution X-band radar data, these are currently only available at experimental sites without large spatial coverage and with short ranges which hamper extrapolation. A future research challenge will be to combine the coarser resolution radar data, which is available at large scale, with the higher-resolution but more local rainfall estimates (Nielsen et al., 2014b). The coarser but larger scale radar data allow estimation of velocity fields and the advection of radar composites, whereas the local higher resolution



estimates allow near-real-time spatial interpolation and dynamic calibration of the stochastic noise model parameters. Additional blending or assimilation with numerical weather prediction models increases the lead time (Liguori and Rico-Ramirez, 2012, 2013; Jensen et al., 2015; Korsholm et al., 2015)

## 4 The value of radar rainfall for urban hydrology

The field of urban hydrology has over the last decade expanded the focus from analysis, design and operation of urban stormwater systems and wastewater treatment plants. Today, the key drivers of research include urban city resilience to hydrological extremes, water and resource recovery, climate change impacts and adaption as well as integration with other city planning and management disciplines, including urban development. This has led to a need for new and more diverse precipitation inputs, both to address the challenges mentioned above, and also because urban hydrology is becoming more

complex with implementation of sustainable stormwater management infrastructure. This increased complexity often implies that the spatial distribution of precipitation becomes even more important in both planning and operation of urban systems, and therefore urban hydrology will require better resolved rainfall products in the future. The current main application fields for radar rainfall in urban hydrology are shown in Table 3. As shown in the table several new application fields have emerged over the last decade. Radar measurements can provide important contributions to these new fields. The

improvements discussed in the previous section have also enhanced the possibility to use radar data in the existing application fields.

### 4.1 General statistical and hydrometeorological characterization of precipitation at urban scale

The long-term analysis of single- or multi-site rain gauges continues to receive substantial attention in the design and analysis of urban water infrastructure, both for quantifying uncertainty and for studying non-stationary behavior (e.g.

Ntegeka and Willems, 2008; Madsen et al., 2009; Willems, 2013a, 2013b; Gregersen et al., 2014). However, the field of assessing future precipitation extremes due to anthropogenic climatic changes is heavily dependent on spatially distributed information on precipitation quantities. Previously, Areal Reduction Factors proved to be sufficient to describe the spatial distribution of extremes (e.g. Sivapalan and Blöschl, 1998 and Vaes et al., 2005), but the *Global Circulation Models (GCM)* used to describe anticipated future climatic changes require spatially distributed precipitation estimates for verification

purposes. Such datasets are therefore set up with historical data with the same spatial resolution as the *GCMs* in order to test the model performance on current climate as a measure of its accuracy of predicting future changes. These datasets are typically still based on point measurements, but there are known shortcomings of this approach, especially in areas with low density of measurement stations (e.g. Haylock et al., 2008; Lenderink, 2010). Radar rainfall data is expected to be able to provide better estimates of precipitation for these gridded datasets. An example of such an application is Kendon et al.

(2014), where a high resolution model (1.5 by 1.5 km grid size) covering part of the UK is validated against a 9 year series of radar rainfall data, because a suitable gridded dataset cannot be constructed based on point measurements. Similar datasets





are being constructed for other regions (Overeem et al., 2009; Thorndahl et al., 2014b; Wright et al., 2014; Berg et al., 2015; Goudenhoofdt and Delobbe, 2016). Climate change models with this high resolution can provide much better physical description of the climatic changes of sub-daily extreme precipitation in high spatial resolution (Tabari et al., 2016) and hence such uses are clearly an emerging field for radar applications in urban hydrology.

As radar data quality improves, it can also be directly used to estimate precipitation extremes, for example in the form of traditional *intensity-duration-frequency* curves (e.g. Overeem et al., 2009b, 2010; Marra and Morin, 2015; Paixao et al., 2015). For small-scale urban applications this requires an understanding of the spatial variability in rainfall at the radar subpixel scale. Recent advances in stochastic space-time rainfall modelling allow the quantification of this subpixel variability explicitely and the generation of ensembles of *IDF* curves at radar subpixel scale which remove the bias in radar

*IDF* curves (Peleg et al., 2016). This can be of major importance for very local estimates of rainfall extremes from radar data.

## 4.2 Re-analysis of damaging extreme events

Re-analysis of extreme events was mentioned in Einfalt et al. (2004) as an important field of application of radar rainfall and a field where good approaches had been developed. The continued development of radars have enabled very accurate re-

analyses of historical events (Jessen et al., 2005; Smith et al., 2013; Yang et al., 2013; Thorndahl et al., 2014a, 2014b; Wright et al., 2014a, 2014b).

The field of distributed 1D-2D hydraulic/hydrological models for urban flood simulations have matured and standard methods have been developed (e.g. Zhou et al., 2012, Henonin et al., 2013). The state of the art described in these papers use point rainfall statistics, mainly because the uncertainty of estimating volume estimates for very high return periods

supersedes the uncertainty related to the internal dynamics of how to distribute this rainfall estimate within the catchment. However, recent studies have also partitioned the contribution of spatial and temporal variability in rainfall to urban flow quantiles, and shown that spatial rainfall variability does matter, especially for high return periods (e.g. Peleg et al., 2016a).

## 4.3 Urban Water Management

The paradigm of using point rainfall data from rain gauges in very high temporal resolution, assuming it to be representative

of an entire urban catchment is challenged by several factors. First, rainfall data from high resolution radar have shown high spatial variability at the intra-urban scale. Moreover, many cities experience substantial development in the form of urban sprawl. This leads to very large cities, where uniform precipitation cannot be assumed, because the catchment size is larger than the spatial representativeness of point precipitation. Hence, there is a far more complex hydrological response from large urban and peri-urban areas compared to smaller urban areas.

Another driver is the climate change adaptation needs of larger cities. Many countries and regions explicitly mention Nature Based Solutions or Sustainable Urban Drainage Systems, as a very important component in this adaptation, including countries and regions such as China, EU, and Australia. These wetlands, rain gardens, soakaways etc. are making the





hydrological response of cities more complex. Hence, there is a need for generating spatially distributed rainfall series in high resolution in space and time. As mentioned in Section 4.1 such series are becoming available in a few places based on radar measurements. Means to develop artificial series based on stochastic properties are being investigated (e.g. Raut et al., 2012 and Sørup et al., 2015), but there is a long way to go before standard procedures have been identified. These

procedures will almost certainly in any case be based on spatially distributed rainfall observations such as radar rainfall observations.

## 4.4 Nowcasting and operational warning

With higher frequency and risk of damage due to heavy rainfall in urban areas as a consequence of climate change and increased urbanization, there is a motivation to develop in developing reliable warning systems, which have a higher level of

detail towards urban hydrology, than traditional numerical weather prediction model forecasts of heavy rainfall, cloud bursts, hurricanes, etc. The evolution in computational power and models enables operational weather models to provide finer resolutions than just a few years ago. However, neither temporal resolution nor spatial resolution are currently fine enough to resolve rainfall sufficiently enough for many urban hydrological applications (e.g. Thorndahl et al., 2016). Furthermore, weather models may still have offsets of tens of kilometers in terms of predicting the exact location a rain cell. This

constitutes a significant problem in applying weather model data for urban hydrological systems, where the location of heavy rainfall is key. For short lead times this problem can to some extent be solved by assimilating radar nowcasts into numerical weather models in order to improve initial conditions of the latter (Jensen et al., 2015). This is however not yet operational, so in order to issue valid urban hydrological warnings, it can be beneficial to have 1) on-line rainfall estimates in high temporal and spatial resolution from radars and potentially also nowcasted data, as well as 2) on-line information on the

current state of the hydrological system, e.g. baseflow, soil saturation, residual storage capacity, etc.

Examples of operational warnings systems based on radar data are: local flood warning systems, systems for emergency planning in case of flooding, warning systems for capacity of receiving waters, etc. Operational warning systems based on radar observations have potentials in rainfall-warnings if radar rainfall estimates exceed a specified threshold (e.g. Einfalt and Luers, 2015) or as hydrological warnings where radar observations (or nowcasts of radar data) are applied as input to an

on-line hydrological model as described above. With regards to the latter there are still rather few applications of operational on-line distributed 2D or 1D-2D flood warning models, since they tend to be too computational expensive to run in real-time (e.g. Thorndahl et al., 2016). Instead simplified, lumped models, or 1D-models are often applied (Bell and Moore, 1998; Sharif et al., 2006; Javier et al., 2007; Smith et al., 2007; Fang et al., 2008; Einfalt et al., 2009; Duncan et al., 2013; Wolfs et al., 2016).

In the literature, there are various examples of real-time operation of urban drainage models, which are applied to warn if flow, water level, combined sewer overflow volume, storage filling, etc. exceeds certain thresholds, e.g. Yuan et al. (1999), Vieux and Bedient (2004a, 2004b), Vieux et al. (2008), Achleitner et al. (2009), Liguori et al. (2012), Liguori and Rico-Ramirez (2012), Schellart et al. (2012a), Dirckx (2013), Thorndahl et al. (2013), Thorndahl and Rasmussen (2013), Löwe et





al. (2014), Schellart et al. (2014), and Löwe et al. (2016). Several of these are pre-operational and have studied the potentials of applying radar data (with or without nowcasting) in real-time prediction of sewer system states.

Simulation of the probabilistic urban rainfall nowcasts in urban drainage models allows probabilistic nowcasts to be obtained of the inundation hazards and risks in urban areas. Ntegeka et al. (2015) have shown how probabilistic urban inundation risk

maps can be obtained by combining STEPS-based rainfall nowcasts with a nested 1D-2D sewer hydraulic and surface inundation model, and a model to assess the damages and social consequences of the urban inundations. Such system, however, only becomes useful for operational management when the uncertainties in the inundation risks can be communicated in a compact and clear way, and when these are informative and manageable by decision makers or the wider public.

## 4.5 Operational real-time control

Model based real-time control of urban drainage systems has evolved significantly during the past decade. Many model based real-time control methods were developed for applications with on-line in-sewer instrumentation or rain gauges for local systems (e.g. Schütze et al., 2004). With advances in estimating spatially distributed rainfall with radars, it is possible to implement real-time control on a much larger scale, e.g. a whole city. By exploiting the spatial variability of rain and

successive unequal local loading of the hydrological systems, novel developed methods aim at utilizing spare capacity systems in order to reduce spills, overflows, flooding, etc. (e.g. Faure and Auchet, 1999; Pfister and Cassar, 1999; Mounce et al., 2014).

Other real-time control applications have been used to estimate the loads on waste water treatment plants in order to reduce spill of untreated waste and stormwater and to optimize treatment processes during rain (Quirmbach and Schultz, 1999;

Fuchs and Beeneken, 2005; Thorndahl et al., 2013; Vezzaro and Grum, 2014a; Kroll et al., 2016). With large linked hydrological systems, centralization of treatment plants in urban areas, advances in model predictions and data, there seem to be a large potential for global predictive control of hydrological systems in cities, which is not yet fully exploited.

## 5   Summary and recommendations

This paper summarized literature findings from the last decade in three key research areas: temporal and spatial radar rainfall

resolution in relation to their use in urban hydrology, radar rainfall data adjustment and quality, and use of radar data for rainfall nowcasting and on-line applications.

In the following, we summarise emerging developments and applications of radar rainfall in urban hydrology that were identified in this review and provide recommendations for future research as well as practical recommendations for the application of radar rainfall in urban hydrology:



1. *Radar resolution*

   A recent and promising development is the installation of X-band polarimetric radars in urban areas, providing high resolution rainfall estimates, typically at or below 1 minute and 100 meters, but with a shorter range than C- and S-band radars. While X-band radar is sensitive to attenuation due to its frequency band, the use of polarimetric signals provides additional parameters insensitive to attenuation, thus solving an important problem associated with X-band radars. While dual polarimetric radars are capable of providing an independent rainfall product, single polarimetric X-band radars on the other hand, require extensive post-processing incorporating data from additional sensors to obtain reliable, high resolution rainfall estimates. In S- and C-band radar networks, high resolution products are starting to be developed, based on for instance shortened pulse lengths. This reveals a transition from use of primarily research radars with high resolution to more commercially produced products from meteorological services focusing also on high resolution for urban hydrological application.

   Where high resolution radar rainfall products are not available, spatial and temporal downscaling (advection interpolation) is applied to obtain higher resolution rainfall estimates, starting from coarse resolution radar products. Downscaling can be based on physical processes or on stochastic principles, the latter being more flexible for including uncertainty, and being less computationally intensive, but also having more difficulty in reproducing the natural, physical structure of storms.

2. *Radar data adjustment and rainfall data quality*

   Radar rainfall estimates suffer from uncertainties associated with variability in drop size distribution, partial beam filling, overshooting, and signal attenuation. One way to reduce these uncertainties is by using polarimetric signals, another way is by reducing distance to the radar, by increasing the density of the radar network. Both require significant investments and in many situations are not foreseen in the near future. This implies that radar data adjustment based on a network of ground stations (rain gauges) will still be required to reduce rainfall data uncertainty. The quality of radar data adjustment in turn depends on the density and quality of the rain gauge network. The optimal temporal integration period or spatial aggregation level for radar adjustment is directly related to the ability of the rain gauge network to capture the temporal and spatial variability of the rain. It is difficult to recommend specific gauge network densities for radar rainfall adjustment since the optimal value will depend on storm type, homogeneity of the rain gauge network, orographic features of the rain, adjustment methods, etc. as well as the specifications of the urban hydrological application.

3. *Nowcasting of rainfall and on-line applications*

   Whereas numerical weather forecast models have too coarse a spatial and temporal resolution for valid forecasts in urban hydrological applications, the use of short-term forecasting (nowcasting) of radar rainfall shows potential in many on-line urban hydrological applications with warning systems or real-time control of urban hydrological systems. Currently, there are some drawbacks with pure radar extrapolation methods in terms of predicting convective rainfall with rapidly evolving storm structure evolution. In order to overcome these problems stochastic



blending of radar rainfall observations/extrapolations with numerical weather prediction models ensembles shows potentials for flash flood/fast hydrological response systems.

Flood warning/pluvial flood warning for small urban catchments based on critical rainfall threshold or flash flood/pluvial flood warning based on real-time urban hydrological modelling are expected to be developed significantly in forthcoming years in order to adapt to climate changes and increased urbanization.

For urban hydrological applications in general, higher resolution and higher accuracy rainfall estimates are beneficial for a better understanding of the hydrological response. Higher accuracy comes with required investments in equipment (X-band radar, polarimetric capability or dense rain gauge network for adjustment) that need to be justifiable either from a research or a societal perspective (higher efficiency of operational control, more accurate early warning). Some general recommendations can be derived from the recent literature as to requirements for radar rainfall resolution: studies have shown that the sensitivity of hydrological response and thus added value of higher resolution rainfall data input increases for smaller catchment size, higher catchment spatial variability, smaller storm size, higher storm variability and higher storm movement velocity. An important consideration here is that accuracy of rainfall estimates generally decreases for higher resolution: accuracy of storm total rainfall is typically much higher than for 5-15 minute rainfall estimates, the same applies for spatial aggregation levels. Conversely, higher rainfall measurement resolution results in higher accuracy of rainfall estimates, than if rainfall estimates are derived from coarse resolutions. In applications, a balance will always be needed between the benefit of higher accuracy and required investment of obtaining such accuracy. In a nowcasting and near real-time context, challenges are even higher, because data correction and adjustment windows are typically short, while false early warnings can have large societal impacts.

Whereas the initial word of the title of the Einfalt et. al. 2004 paper (*Towards a roadmap for use of radar rainfall data in urban drainage*) suggested a progression, this paper has demonstrated that radar rainfall data presently is applied in many operational systems and that radar has become an invaluable source of data along with other types of observations applied in urban hydrology. This said, there is still research to be conducted in order to improve radar rainfall estimates, e.g. especially with regard to dual-polarimetry, uncertainty assessment and generation of realistic ensemble forecasts, combined radar and weather model rainfall products, and real-time applications.

## 6   Author contribution

S. Thorndahl coordinated the joined collaboration and developed the greater part of the manuscript with partial contributions from other co-authors on: radar uncertainties and data quality (T. Einfalt); nowcasting and real-time applications (P. Willems); spatial and temporal resolution (J.E. Nielsen); X-band polarimetry, summary and recommendations (M.-C.t



Veldhuis); Off-line applications and future outlooks (K. Arnbjerg-Nielsen); technical radar specifications (M. Rasmussen); and radar rainfall extremes and proofreading (P. Molnar).

## 7 Acknowledgements

We acknowledge the International Working Group on Urban Rainfall (IGUR) under the IWA/IAHR Joint Committee on Urban Drainage for providing the network supporting the collaboration in writing this paper. S. Thorndahl acknowledges Damian Murla Tuyls from Department of Civil Engineering, Aalborg University for constructive ideas and proofreading.

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




**Tables**

**Table 1. Most cited papers within radar and urban drainage**

| | Paper title | Citations Scopus[1] | Citations Web of Science[1] |
|---|---|---|---|
| Einfalt et al. (2004) | Towards a roadmap for use of radar rainfall data in urban drainage | 88 | 75 |
| Smith et al. (2002) | The Regional Hydrology of Extreme Floods in an Urbanizing Drainage Basin | 84 | 76 |
| Schilling (1991) | Rainfall data for urban hydrology: what do we need? | 76 | 71 |
| [1] per 01 October 2016 | | | |

5   **Table 2. Typical operating resolutions and maximum ranges for different types of weather radars used in hydrological applications.**

| | X-band | C-band | S-band |
|---|---|---|---|
| Spatial resolution | 100-1000 m | 250-2000 m | 1000-4000 m |
| Temporal resolution | 1-5 min | 5-10 min | 10-15 min |
| Maximum quantitative range | 30-60 km | 100-130 km | 100-200 km |

**Table 3. Application fields for radar rainfall in urban hydrology. Applications that have emerged significantly since Einfalt et al. (2004) are marked with bold. Numbers in parenthesis indicate which sub-section discusses the particular application.**

| Off-line applications | On-line applications |
|---|---|
| - General statistical and hydrometeorological characterization of precipitation at urban scale (4.1)<br>  - Present climate<br>  - **Extremes**<br>  - **Future climate**<br><br>- Re-analysis of damaging extreme events (4.2)<br>  - Insurance claims<br>  - **Hydrological re-analysis of flood events**<br>  - **Distributed hydrological modelling for flood risk assessment**<br><br>- Urban water management (4.3)<br>  - Design of basins and pipes<br>  - **Resilience and livability measures** | - **Nowcasting and operational warning (4.4)**<br>  - Severe rainfall warning<br>  - **Flow/flood warning based on on-line hydrological models**<br><br>- **Operational real-time control of hydrological systems (4.5)**<br>  - Nowcasting<br>  - Real-time hydrological models with data assimilation<br>  - **Scenario/ensemble modelling for on-line evaluation of control strategies** |



**Figures**

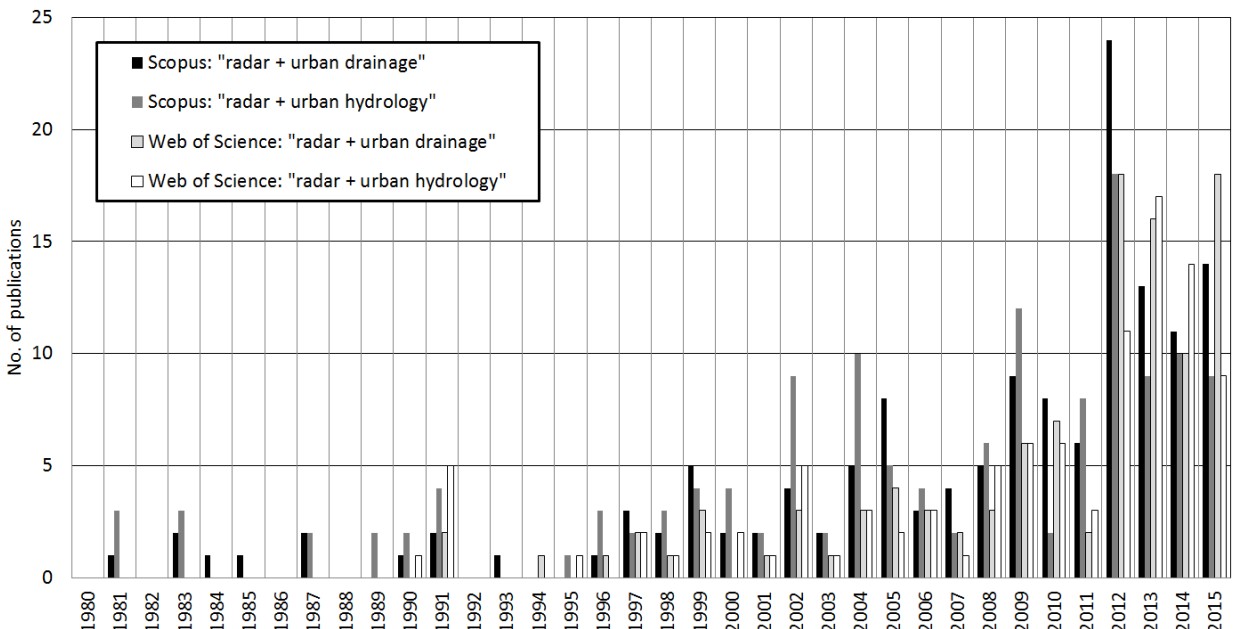

**Figure 1: Scopus and Web of Science documents under search strings "*radar + urban drainage*" and "*radar + urban hydrology*".**





**Figure 2: Example of radar reflectivity in four different cartesian spatial resolutions over Aalborg, Denmark (Lat: 57.05, Lon: 9.92). The radar data is acquired with a Furuno WR-2100 dual-polarimetric X-band radar (Nielsen et al., 2015) in 1 min. temporal resolution at 16:20:00 UTC on July 25, 2016. Black circles are rain gauges of the Danish Water Pollution Committee network.**

