# Peer review of "Weather radar rainfall data in urban hydrology"

_Hydrology and Earth System Sciences, 2016_

## Referee Comment (RC1) · D. Wright (Referee) · 2 Dec 2016

The authors present a review of weather radar technological and methodological advances in light of more than a decade of progress since the well-known Einfalt et al. (2004) review. The paper is a pleasure to read and represents a useful update on the state of the knowledge. I have only minor comments, mainly grammatical, that the authors should address prior to publication.

Pg. 1 Line 14: delete the comma after "hydrology"

Introduction: I do think it would be useful to mention what "urban hydrology" means, though perhaps the authors think it is self-evident. Later in the paper, a number of specific application topics are mentioned, but perhaps a brief list belongs in the introduction.

Pg.2 Line 24: delete "of" after "many"

Pg.2 line 31: delete comma after "models"

Pg.3 lines 3-4: this sentence structure is awkward

Pg.3 lines 10-12: This may be nitpicking, but the principles of precipitation measurement using radar should not change-at least if "principles" refers to the underlying physics of both radar and precipitation. I'd suggest changing this sentence.

Pg.5 line 5: In my mind, this should say "down to a minimum of 500 m" rather than "up to a maximum of 500 m."

Pg.6 line 17-18: I understand this "fishbone" idea, but if the authors have a figure available that demonstrates it, they could consider including it in the paper.

Pg.6 line 19: See later comment on usage of term "commercial radar rainfall products"

Pg.9 line 31-Pg.10 line 1: Wright et al. (JAWRA 2014) also examined the role of gage density in MFB estimation.

Sections 3.2.2 and 3.2.3: It seems strange that these are separate sections-the content of section 3.2.3 seems to naturally fit within the scope of Section 3.2.2. I am also surprised that range effects don't appear in this discussion, and possible solutions such as approaches based on the vertical reflectivity profile. In addition, it is perhaps worth noting that MFB has an implicit range adjustment feature, in that, at least for storms that don't cover a large portion of the radar coverage, the gages reporting positive rain will be spatially close to each other, i.e. at similar distance from the radar, and thus the computed MFB will be in some sense "tailored" to compensate for range dependent bias. This could be worth mentioning, as MFB is sometimes viewed as being overly simplistic when in fact, for this reason and others, it works quite well.

Section 3.2.4: I object to the wording "commercial radar rainfall products." Perhaps

"commercial" has a different implication in Europe but in North America it implies that the product would be available for purchase from some private-sector. While such products certainly exist, the authors refer to products produced by government agencies that, at least in the United States, are available free of charge.

Pg. 13 line 13: Consider changing "cases" to "cells" or "elements"

Section 4.1: The first paragraph of this section is at times hard to follow. I'm not sure what the sentence on pg.14 lines 20-22 is trying to say. Is it that climate projections from GCMs are "spatially distributed"? Or that we need to understand the spatial nature of precipitation extremes in a changing climate? Please reword as needed. If the intended meaning is that GCM outputs are spatially distributed, I would argue that this isn't the case, due to their coarse resolution. Instead, I would argue that they are "spatially averaged," and so the relevant methods needed to use such information revolve around using radar (or other methods) to disaggregate these coarse spatial averages to finer scales. I'm also having a hard time understand the sentences on pg.14 lines 22-29, regarding the connection between Area Reduction Factors (ARF) and GCM outputs, point-based historical data, etc. These are all relevant issues, but the connections need to be clearer. If the authors wish to mention work related to ARF estimation using radar, they could consider the work of Durrans et al., (2002) and Wright et al. (2014).

Pg.15 line 1: Multiple Wright et al. 2014 papers are included in the bibliography-which is referred to here?

Section 4.4: I think it is worth mentioning past work and future potential for assimilation of radar data into short-term numerical weather forecasts. Great potential here, I recently reviewed a paper (not yet published) with an urban application using NCAR's DART system that showed excellent results in an urban setting.

The authors don't say a lot in the review about the future: data assimilation, refinement of dual-polarization algorithms, phased array technology, etc. Consider including a

brief mention and references.

The authors could mention more specific research efforts, such as the CASA network (http://www.casa.umass.edu/index.php).

References: Durrans, S.R., Julian, L.T., Yekta, M., 2002. Estimation of Depth-Area Relationships using Radar-Rainfall Data. J. Hydrol. Eng. 7, 356–367.

Wright, D.B., Smith, J.A., Baeck, M.L., 2014. Critical Examination of Area Reduction Factors. J. Hydrol. Eng. 19, 769–776.

---

## Author Comment (AC1) · 11 Dec 2016

On behalf of the authors I would like to thank Daniel Wright for his review of the paper. We are happy that he thinks the paper is a pleasure to read and useful. In a revision of the paper we will address the comments and constructive additions that Wright suggested.

best regards Søren Thorndahl
* * *

---

## Referee Comment (RC2) · R. Uijlenhoet (Referee) · 17 Jan 2017

Attached please find an annotated manuscript with a significant number (91) of comments and suggestions. The most important of these are:

GENERAL REMARK

- This review paper, although dealing with relevant issues, has become quite lengthy, sometimes reading more as a report than as a scientific paper. Would it be possible to significantly reduce the length of the text, using the saved space to add one or two examples of urban hydrological applications of weather radar, which are currently lacking?

SPECIFIC REMARKS

- P.2, "the significant growth [in the number of papers]": How does the growth in this specific subject area compare to the overall growth of papers in the mentioned

databases? In other words, is the reported growth merely a reflection of the overall increase in the number of publications, or is the relative proportion of papers in this subject area increasing with respect to other topics?

- P.3, "journal papers such as": See also: Delrieu, G., I. Braud, A. Berne, M. Borga, B. Boudevillain, F. Fabry, J. Freer, E. Gaume, E. Nakakita, A. Seed, P. Tabary, and R. Uijlenhoet, 2009: Weather radar and hydrology. Adv. Water Resour., 32, 969–974, doi:10.1016/j.advwatres.2009.03.006.

- P.5, "the radial resolution (or range resolution) is a function of the pulse and wavelength": In principle, the range resolution is equal to half the pulse length, independent of wavelength. See any radar meteorology textbook, such as Louis Battan's classic "Radar Observation of the Atmosphere" (University of Chicago Press, 1973).

- P.5, "each radar scanline is subdivided into a fixed/selected number of range bins": For pulsed radars, the number of range bins is determined by the ratio of the maximum unambiguous range and the range resolution (i.e. half the pulse length). For frequency modulated - continuous wave (FM-CW) radars, the number of range bins is typically fixed at some power of 2 (e.g. 512).

- P.5, "Small, local X-band radars with non-parabolic antennas": Many X-band rainfall radars still employ parabolic dish antennas. The angular resolution of a parabolic dish antenna is proportional to lambda / D, where lambda is the employed radar wavelength and D the antenna diameter. In other words, the larger the antenna (at a fixed wavelength), the more focused the beam. On the other hand, for a given antenna size, the larger the wavelength, the less focused the radar beam. X-band is about 3 cm, C-band 5-6 cm and S-band $\sim$10 cm. Hence, for a given antenna size, the beam width at X-band is $\sim$3 times smaller than at S-band. Or, for an X-band radar the antenna can be 3 times smaller than at S-band to achieve the same angular resolution.

- P.5, "larger opening angles": In some urban hydrological studies, refurbished ship radars are being used as rain radars. Such radars employ the typical horizontal antenna shapes we know from ships. Such antennas produce so-called fan beams, with a small angular resolution in the horizontal direction, but quite a large angular resolution in the vertical direction. In other words, the shape of the radar beam is highly asymmetrical in this case, effectively integrating rainfall over a large vertical distance.

- P.5, "X-band radars function with both higher spatial and temporal resolution": This is typically because X-band radars require smaller antennas than C- and S-band radars to achieve the same angular resolution. Smaller antennas are much easier to rotate quickly, thereby increasing the effective temporal resolution with which the rainfall field is being sampled.

- P.5, "different volume scans": Rather than "different volume scans" I would say "scans at different elevation angles" (comprising one volume scan).

- P.6, "methods to interpolate between radar images": The classic reference on this topic is: Frederic Fabry, Aldo Bellon, Mike R. Duncan, Geoffrey L. Austin (1994): High resolution rainfall measurements by radar for very small basins: the sampling problem reexamined. J. Hydrol. 161 (1-4), 415-428.

- P.7, "a relation between the temporal and the spatial resolution": Extrapolating the results of Van de Beek et al. (2012) leads to $r = 5\ t\ \hat{}\ 0.3$ for summer conditions in the Netherlands. See: Van de Beek, C.Z., H. Leijnse, P.J.J.F. Torfs, and R. Uijlenhoet, 2012: Seasonal semivariance of Dutch rainfall at hourly to daily scales. Adv. Water Resour., 45, 76–85, doi:10.1016/j.advwatres.2012.03.023.

- P.8, "the power-law parameters will vary with the DSD": I would say: "... vary with the DSD shape". If they would vary with the DSD they would vary continuously and that is not the case. However, Z-R relations do vary if the general shape of the DSD changes (e.g. from exponential to gamma).

- P.9, "under the assumption that the radar field has a homogeneous DSD": This is not the only tacit assumption. Another one is that there are no systematic range effects in

the radar rainfall retrievals (e.g. due to an increasing beam height in combination with the vertical reflectivity profile, or due to attenuation).

- P.9, "1 gauge per 10-20 km2 for urban areas": Berne et al. (2004) also provide numbers concerning the required spatial rainfall resolution for urban hydrological applications. You have referred to this before.

- P.10, "adjusted or merged with rain gauge network data": However, the employed adjustment or merging methods are often quite straightforward, consisting of a combination of a mean field bias correction and a range-dependent correction. This approach is applied a.o. at SMHI (see the work of Daniel Michelson) and KNMI (the work of Iwan Holleman). I think UKMO uses a similar approach. In other words, I have the impression that geostatistical merging methods are largely limited to the academic community.

- P.12, "leaving accurate radar rainfall adjustment less crucial": I am not so sure about this. Even a small but persistent bias, if lasting long enough, can be detrimental for hydrological simulations (e.g. rainfall-runoff modeling), even if the model is uncertain. See e.g.:

Brauer, C.C., A. Overeem, H. Leijnse, and R. Uijlenhoet, 2016: The effect of differences between rainfall measurement techniques on groundwater and discharge simulations in a lowland catchment. Hydrol. Proc., 30, 3885–3900, doi:10.1002/hyp.10898.

- P.12: Classical references on this topic are:

Austin, G.L. and Bellon, A., 1974. The use of digital weather radar records for short-term precipitation forcasting. Q. J. R. Meteorol. Soc., 100: 658-664.

Einfalt, T., Denoeux, T. and Jacquet, G., 1990. A radar rainfall forecasting method designed for hydrological purposes. J. Hydrol., 14:229-244.

- P.13: I would also add SBMcast:

Berenguer, M., C. Corral, R. Sánchez-Diezma, D. Sempere-Torres, 2005: Hydrological validation of a radar-based nowcasting technique. J. Hydrometeor, 6, 532–549. doi: http://dx.doi.org/10.1175/JHM433.1.

Berenguer, M., D. Sempere-Torres, and G.S. Pegram, 2011: SBMcast – An ensemble nowcasting technique to assess the uncertainty in rainfall forecasts by Lagrangian extrapolation. J. Hydrol., 404, 226–240, doi:10.1016/j.jhydrol.2011.04.033.

- P.16: Is it really necessary to mention both "frequency" and "risk", or would only "risk" suffice?

- P.10, 18: I would use "operational" rather than "commercially produced". Many national meteorological services are not commercial at all.

Remko Uijlenhoet

Please also note the supplement to this comment:
http://www.hydrol-earth-syst-sci-discuss.net/hess-2016-517/hess-2016-517-RC2-supplement.pdf

**Supplement:**

[revised manuscript text omitted]

  - Present climate
  - **Extremes**
  - **Future climate**

- Re-analysis of damaging extreme events (4.2)
  - Insurance claims
  - **Hydrological re-analysis of flood events**
  - **Distributed hydrological modelling for flood risk assessment**

- Urban water management (4.3)
  - Design of basins and pipes
  - **Resilience and livability measures** | - **Nowcasting and operational warning (4.4)**
  - Severe rainfall warning
  - **Flow/flood warning based on on-line hydrological models**

- **Operational real-time control of hydrological systems (4.5)**
  - Nowcasting
  - Real-time hydrological models with data assimilation
  - **Scenario/ensemble modelling for on-line evaluation of control strategies** |

[Figure]

[Figure]

**Figures**

[Figure]

**Figure 1: Scopus and Web of Science documents under search strings "*radar + urban drainage*" and "*radar + urban hydrology*".**

[Figure]

[Figure]

**Figure 2:** Example of radar reflectivity in four different cartesian spatial resolutions over Aalborg, Denmark (Lat: 57.05, Lon: 9.92). The radar data is acquired with a Furuno WR-2100 dual-polarimetric X-band radar (Nielsen et al., 2015) in 1 min. temporal resolution at 16:20:00 UTC on July 25, 2016. Black circles are rain gauges of the Danish Water Pollution Committee network.

---

## Author Comment (AC2) · 27 Jan 2017

On behalf of the authors I would like to thank Remko Uijlenhoet for his profound review of the paper. We are grateful for minor comments, finding of spelling and grammatical errors, need for clarifications, etc., as well as more detailed comments/suggestions. Especially, section 3.1.2 on spatial resolution can be improved significantly by Uijlenhoet's comments, and we acknowledge that many issues in this section could have been explained better in the first submission of the paper. Moreover, we appreciate that Uijlenhoet have suggested several additional references which are indeed relevant to cite.

With regards to the general remark on the length of the paper. We agree that the paper have become rather large, and we've been struggling with this during the writing process. At an earlier stage we had a draft with more general radar physics and general uncertainties in estimating rain with radar – which are indeed relevant to include, but

which also can be found in many other papers. We therefore decided to only include what we, subjectively, find relevant for direct applications within urban hydrology. With a reduction of the manuscript, we are afraid the common thread throughout the paper might be lost and we will be forced to refer to other papers for clarifications on different topics. We can try to remove some less important paragraphs here and there, but it will be difficult to remove whole sections without compromising on the clarity and continuity of the paper.

With regards to the suggestion to add one or two examples on urban hydrological applications of weather radar. This is indeed a good idea, which was also discussed by the authors during the writing process. Although one or two examples might be insufficient to cover all urban hydrological applications of weather radar, it could be relevant to include examples where we think application of radar rainfall data might contribute the most to the field. In line with the example on spatial resolution, we could e.g. include an example of a flow forecast based on ensemble radar nowcasts with different lead times. Adding a figure or two might also enhance the readability of the paper.

On behalf of the authors Søren Thorndahl

---

## Author Response (AR1)

**Reply to editor and reviewers**

Dear editor

Please find details below on how we have implemented different reviewer suggestions in the manuscript. Minor corrections, typos, etc. are only corrected in the document (see track changes). Replies are marked with yellow.

On behalf of the authors
Søren Thorndahl

**Handling Editor: Chris Onof suggests a minor revision with the comments:**
Comments to the Author:
The reviews are very positive. I agree with Remko Uijlenhoet's comment about the problem with the length of the paper. However, as you correctly point out in the reply, it is not easy to take out whole sections without affecting the flow of the logic of the paper. I suggest therefore that you consider whether there are a few paragraphs containing less important material that could be taken out without affecting the structure of the whole.
Once this is done, and other minor corrections are carried out, this paper will be publishable

We have restructured parts of the paper and deleted paragraphs here end there in order to reduce the length of the paper. Moreover we have deleted parts of section 2 and added some parts to the introduction and the state of the art section. Some new references have been added and some sentences have been rephrased in order to make them shorter and clearer to the point.

**Comments from R. Uijlenhoet**
GENERAL REMARK
- This review paper, although dealing with relevant issues, has become quite lengthy, sometimes reading more as a report than as a scientific paper. Would it be possible to significantly reduce the length of the text, using the saved space to add one or two examples of urban hydrological applications of weather radar, which are currently lacking?

See comment to editor.

SPECIFIC REMARKS
- P.2, "the significant growth [in the number of papers]": How does the growth in this specific subject area compare to the overall growth of papers in the mentioned databases? In other words, is the reported growth merely a reflection of the overall increase in the number of publications, or is the relative proportion of papers in this subject area increasing with respect to other topics?

A paragraph has been added comparing the growth in hydrology in general to the growth in radar related publications.

- P.3, "journal papers such as": See also: Delrieu, G., I. Braud, A. Berne, M. Borga, B. Boudevillain, F. Fabry, J. Freer, E. Gaume, E. Nakakita, A. Seed, P. Tabary, and R. Uijlenhoet, 2009: Weather radar and hydrology. Adv. Water Resour., 32, 969–974,

doi:10.1016/j.advwatres.2009.03.006.

==Reference added==

- P.5, "the radial resolution (or range resolution) is a function of the pulse and wavelength":
In principle, the range resolution is equal to half the pulse length, independent
of wavelength. See any radar meteorology textbook, such as Louis Battan's classic
"Radar Observation of the Atmosphere" (University of Chicago Press, 1973).

==A paragraph is added and ref is included==

You probably mean the so-called maximum unambiguous range?
- P.5, "each radar scanline is subdivided into a fixed/selected number of range bins":
For pulsed radars, the number of range bins is determined by the ratio of the maximum
unambiguous range and the range resolution (i.e. half the pulse length). For frequency
modulated - continuous wave (FM-CW) radars, the number of range bins is typically
fixed at some power of 2 (e.g. 512).

==Clarification has been added, but the point on FM-CW is omitted.==

- P.5, "Small, local X-band radars with non-parabolic antennas": Many X-band rainfall
radars still employ parabolic dish antennas. The angular resolution of a parabolic dish
antenna is proportional to lambda / D, where lambda is the employed radar wavelength
and D the antenna diameter. In other words, the larger the antenna (at a fixed wavelength),
the more focused the beam. On the other hand, for a given antenna size, the
larger the wavelength, the less focused the radar beam. X-band is about 3 cm, C-band
5-6 cm and S-band _10 cm. Hence, for a given antenna size, the beam width at Xband
is _3 times smaller than at S-band. Or, for an X-band radar the antenna can be
3 times smaller than at S-band to achieve the same angular resolution.

==Good point. We were addressing this point only to ship-radars which have been applied fore some applications within urban hydrology. I think it will be too much to include more detailed information here.==

- P.5, "larger opening angles": In some urban hydrological studies, refurbished ship
radars are being used as rain radars. Such radars employ the typical horizontal antenna shapes we
know from ships. Such antennas produce so-called fan beams, with
a small angular resolution in the horizontal direction, but quite a large angular resolution
in the vertical direction. In other words, the shape of the radar beam is highly
asymmetrical in this case, effectively integrating rainfall over a large vertical distance.

==Yes. These refs are all related to these types of radars. Point on the large vertical distance has been added==

- P.5, "X-band radars function with both higher spatial and temporal resolution": This is
typically because X-band radars require smaller antennas than C- and S-band radars
to achieve the same angular resolution. Smaller antennas are much easier to rotate
quickly, thereby increasing the effective temporal resolution with which the rainfall field

is being sampled.

Good point. This has been added to the text

- P.5, "different volume scans": Rather than "different volume scans" I would say "scans at different elevation angles" (comprising one volume scan).

Corrected

P.6, "methods to interpolate between radar images": The classic reference on this topic is: Frederic Fabry, Aldo Bellon, Mike R. Duncan, Geoffrey L. Austin (1994): High resolution rainfall measurements by radar for very small basins: the sampling problem reexamined. J. Hydrol. 161 (1-4), 415-428

This reference has been added

- P.7, "a relation between the temporal and the spatial resolution": Extrapolating the results of Van de Beek et al. (2012) leads to r = 5 t ^ 0.3 for summer conditions in the Netherlands. See: Van de Beek, C.Z., H. Leijnse, P.J.J.F. Torfs, and R. Uijlenhoet, 2012: Seasonal semivariance of Dutch rainfall at hourly to daily scales. Adv. Water Resour., 45, 76–85, doi:10.1016/j.advwatres.2012.03.023.

This point has been added

- P.8, "the power-law parameters will vary with the DSD": I would say: "... vary with the DSD shape". If they would vary with the DSD they would vary continuously and that is not the case. However, Z-R relations do vary if the general shape of the DSD changes (e.g. from exponential to gamma).

Corrected

- P.9, "under the assumption that the radar field has a homogeneous DSD": This is not the only tacit assumption. Another one is that there are no systematic range effects in the radar rainfall retrievals (e.g. due to an increasing beam height in combination with the vertical reflectivity profile, or due to attenuation).

Good point, this is added

- P.9, "1 gauge per 10-20 km2 for urban areas": Berne et al. (2004) also provide numbers concerning the required spatial rainfall resolution for urban hydrological applications. You have referred to this before.

Yes – but this is related to the number of required gauges to perform adjustment – so I don't think the Berne et al (2004) is relevant here.

- P.10, "adjusted or merged with rain gauge network data": However, the employed adjustment or merging methods are often quite straightforward, consisting of a combination of a mean field bias correction and a range-dependent correction. This approach

is applied a.o. at SMHI (see the work of Daniel Michelson) and KNMI (the work of Iwan Holleman). I think UKMO uses a similar approach. In other words, I have the impression that geostatistical merging methods are largely limited to the academic community.

Good point. I added the following: The majority of operational products are based on rather simple range and MFB approaches as described in section 3.2.2 (e.g. Gjertsen et al., 2004).

- P.12, "leaving accurate radar rainfall adjustment less crucial": I am not so sure about this. Even a small but persistent bias, if lasting long enough, can be detrimental for hydrological simulations (e.g. rainfall-runoff modeling), even if the model is uncertain. See e.g.:
Brauer, C.C., A. Overeem, H. Leijnse, and R. Uijlenhoet, 2016: The effect of differences between rainfall measurement techniques on groundwater and discharge simulations in a lowland catchment. Hydrol. Proc., 30, 3885–3900, doi:10.1002/hyp.10898.

The point is here that if you calibrate your hydrological model against flow, water level, obs. etc. a small defect on the rainfall input might blend in with other uncertainties.

- P.12: Classical references on this topic are:
Austin, G.L. and Bellon, A., 1974. The use of digital weather radar records for shortterm precipitation forcasting. Q. J. R. Meteorol. Soc., 100: 658-664.
Einfalt, T., Denoeux, T. and Jacquet, G., 1990. A radar rainfall forecasting method designed for hydrological purposes. J. Hydrol., 14:229-244.

Added

- P.13: I would also add SBMcast:

Berenguer, M., C. Corral, R. Sánchez-Diezma, D. Sempere-Torres, 2005: Hydrological validation of a radar-based nowcasting technique. *J. Hydrometeor*, 6, 532–549. doi: http://dx.doi.org/10.1175/JHM433.1.

Berenguer, M., D. Sempere-Torres, and G.S. Pegram, 2011: SBMcast – An ensemble nowcasting technique to assess the uncertainty in rainfall forecasts by Lagrangian extrapolation. *J. Hydrol.*, 404, 226–240, doi:10.1016/j.jhydrol.2011.04.033.

added

- P.16: Is it really necessary to mention both "frequency" and "risk", or would only "risk" suffice?

Rephased

- P.10, 18: I would use "operational" rather than "commercially produced". Many national meteorological services are not commercial at all.

Good point.We will use "operational"

**Comments from Dan Wright**
Introduction: I do think it would be useful to mention what "urban hydrology" means, though perhaps the authors think it is self-evident. Later in the paper, a number of specific application topics are mentioned, but perhaps a brief list belongs in the introduction.

The following is added:

Where Einfalt et al. (2004) used the term "urban drainage" we extend the terminology to "urban hydrology". Thereby, we do not only encounter design, analysis, and management of urban drainage system, but also urban hydrological modelling/prediction as well as management of and interaction between different parts of the whole urban water cycle, i.e. urban drainage systems, flood prone areas, rivers and streams, ground water, etc.

Pg.6 line 17-18: I understand this "fishbone" idea, but if the authors have a figure available that demonstrates it, they could consider including it in the paper.

I don't think we have the space in the paper to include a figure relating to this term but we have alaborated the description

Pg.9 line 31-Pg.10 line 1: Wright et al. (JAWRA 2014) also examined the role of gage density in MFB estimation.

A paragraph including this reference is added

Sections 3.2.2 and 3.2.3: It seems strange that these are separate sections-the content of section 3.2.3 seems to naturally fit within the scope of Section 3.2.2. I am also surprised that range effects don't appear in this discussion, and possible solutions such as approaches based on the vertical reflectivity profile. In addition, it is perhaps worth noting that MFB has an implicit range adjustment feature, in that, at least for storms that don't cover a large portion of the radar coverage, the gages reporting positive rain will be spatially close to each other, i.e. at similar distance from the radar, and thus the computed MFB will be in some sense "tailored" to compensate for range dependent bias. This could be worth mentioning, as MFB is sometimes viewed as being overly simplistic when in fact, for this reason and others, it works quite well.

Sections are joined and issues with MFB is added and a paragraph is added to the conclusion:

"Conventional MFB adjustment has an implicit range adjustment feature, in that, at least for storms that do not cover a large portion of the radar coverage, the gauges reporting positive rain will be within a close distance to each other and at similar distance from the radar, and thus the computed MFB will be in some sense compensate for range dependent bias. "

"In many studies simple mean field bias adjustment between radar and rain gauges has proven sufficient and robust which is probably also the reason that this method is applied in many operational systems. At present, the more advanced geostatistical approached to bias adjustment are mostly applied within the research community."

Section 3.2.4: I object to the wording "commercial radar rainfall products." Perhaps "commercial" has a different implication in Europe but in North America it implies that the product would be available for purchase from some private-sector. While such products certainly exist, the authors refer to products produced by government agencies that, at least in the United States, are available free of charge.

This has been changed. See suggestion from other reviewer

Section 4.1: The first paragraph of this section is at times hard to follow. I'm not sure what the sentence on pg.14 lines 20-22 is trying to say. Is it that climate projections from GCMs are "spatially distributed"? Or that we need to understand the spatial nature of precipitation extremes in a changing climate? Please reword as needed. If the intended meaning is that GCM outputs are spatially distributed, I would argue that this isn't the case, due to their coarse resolution. Instead, I would argue that they are "spatially averaged," and so the relevant methods needed to use such information revolve around using radar (or other methods) to disaggregate these coarse spatial averages to finer scales. I'm also having a hard time understand the sentences on pg.14 lines 22-29, regarding the connection between Area Reduction Factors (ARF) and GCM outputs, point-based historical data, etc. These are all relevant issues, but the connections need to be clearer. If the authors wish to mention work related to ARF estimation using radar, they could consider the work of Durrans et al., (2002) and Wright et al. (2014).

We have rephrased the whole section!

Section 4.4: I think it is worth mentioning past work and future potential for assimilation of radar data into short-term numerical weather forecasts. Great potential here, I recently reviewed a paper (not yet published) with an urban application using NCAR's DART system that showed excellent results in an urban setting.

More refs are added

The authors don't say a lot in the review about the future: data assimilation, refinement of dual-polarization algorithms, phased array technology, etc. Consider including a brief mention and references.

This is actually some of the issues we discarded in order to shorten the paper.

[revised manuscript text omitted]

  - Present climate
  - **Extremes**
  - **Future climate**

- Re-analysis of damaging extreme events (4.2)
  - Insurance claims
  - **Hydrological re-analysis of flood events**
  - **Distributed hydrological modelling for flood risk assessment**

- Urban water management (4.3)
  - Design of basins and pipes
  - **Resilience and livability measures** | - **Nowcasting and operational warning (4.4)**
  - Severe rainfall warning
  - **Flow/flood warning based on on-line hydrological models**

- **Operational real-time control of hydrological systems (4.5)**
  - Nowcasting
  - Real-time hydrological models with data assimilation
  - **Scenario/ensemble modelling for on-line evaluation of control strategies** |

Slettet: ¶

[Figure]

[Figure]

Slettet:

**Figure 1: Scopus and Web of Science documents under search strings "*radar + urban drainage*" and "*radar + urban hydrology*".**

Slettet: ¶

[Figure]

**Figure 2: Example of radar reflectivity in four different cartesian spatial resolutions over Aalborg, Denmark (Lat: 57.05, Lon: 9.92). The radar data is acquired with a Furuno WR-2100 dual-polarimetric X-band radar (Nielsen et al., 2015) in 1 min. temporal resolution at 16:20:00 UTC on July 25, 2016. Black circles are rain gauges of the Danish Water Pollution Committee network.**

Slettet: ¶

[Figure]

**Figure 3: Examples of runoff forecast prediction in an urban drainage system in Frejlev, Denmark using a radar ensemble nowcast algorithm (Jensen et al., n.d.) An ensemble of 300 nowcasts, a deterministic nowcast, and observed radar data are applied as inputs to an urban drainage model (Thorndahl et al., 2006, Thorndahl and Rasmussen 2013) covering an area of 0.8 km2 with an impervious area of approx. 40 %. Six radar pixels with a 2x2 km2 resolution cover the area. The radar is operated by the Danish Meteorological Institute. Maximum observed rainfall intensities are 6 mm/h and the observed accumulated rainfall is 11.2 mm. It is evident that there is a significant increase in the ensemble spread as a function of the different shown forecast lead times of 10, 30 and 60 minutes, respectively.**

Slettet: ¶